# Clicker Training Mice for Improved Compliance in the Catwalk Test

**DOI:** 10.3390/ani12243545

**Published:** 2022-12-15

**Authors:** Jana Dickmann, Fernando Gonzalez-Uarquin, Sandra Reichel, Dorothea Pichl, Konstantin Radyushkin, Jan Baumgart, Nadine Baumgart

**Affiliations:** Translational Animal Research Center, University Medical Center of the Johannes Gutenberg University, 55131 Mainz, Germany

**Keywords:** clicker training, 3Rs, CatWalk test, refinement, welfare

## Abstract

**Simple Summary:**

Refinement-oriented research remains essential for animal welfare and data reproducibility. When evaluating mouse locomotion, the implementation of the CatWalk XT is helpful for gait assessment, but its application requires eliciting movement of the animals across the corridor, usually by forcing them with unpleasant stimuli. In this study, we tested the efficacy of clicker training to increase performance with the CatWalk test while assessing behavioral changes in the Open Field and Elevated Plus Maze to address the well-being of trained and untrained mice. Clicker training improved running speed on the CatWalk for both sexes. Interestingly, clicker training appeared to reduce anxiety and improve general well-being parameters in the Open Field and the Elevated Plus Maze tests to a greater extent in females. We conclude that clicker training enhances the performance of mice on the CatWalk and is a promising alternative for welfare improvement.

**Abstract:**

The CatWalk test relies on the run of mice across the platform to measure a constant speed with low variation. Mice usually require a stimulus to walk to the end of the catwalk. However, such stimuli are usually aversive and can impair welfare. Positive reinforcement training of laboratory animals is a thriving tool for refinement and contributes to meeting the demands instituted by Directive 2010/63/EU. We have already demonstrated the positive effects of clicker training. In this study, we trained male and female mice to complete the CatWalk protocol while assessing the effects of training on their well-being (Open Filed and Elevated Plus Maze). In the CatWalk test, we observed that clicker training improved the running speed of the mice. In addition, clicker training reduced the number of runs required by mice, which was more pronounced in males. Clicker training lowered anxiety-like behaviors in our mice, especially in females, where a significant difference was observed between trained and untrained ones. Based on our findings, we hypothesize that clicker training is an effective tool to motivate mice and increase performance on the CatWalk test without potentially impairing their welfare (e.g., by puffing them).

## 1. Introduction

Brain damage of different natures—trauma, stroke, degenerative diseases, and genetic manipulations—has often caused an impairment of motor functions. Quantitative measurement of locomotion is an essential method to better understand the mechanisms of this impairment and, most importantly, to evaluate functional recovery after treatment. The Noldus CatWalk XT is a computer-assisted gait analysis setup that allows rapid and objective quantification of many gait parameters in laboratory rodents [1,2,3,4]. The CatWalk XT system detects actual footprints by video recording the animal from below while it traverses a glass plate. Animal compliance is a critical factor for the success of gait analysis. Individual animals must run on a glass plate at the highest possible speed. Only in this case could moderate and subtle deficiencies be appropriately quantified. Moreover, for successful analysis, an animal must maintain its speed constant, that is, with low variation. In this way, a standard practice to promote the movement of mice along the CatWalk includes using air puffs [5], which makes the animals anxious and thus affects their welfare. In addition, it takes a considerable amount of time for the experimenter to make each individual comply to complete the test. If compliance cannot be achieved for all animals, this increases the number of experimental animals required.

Recently, Noldus company has offered an alternative approach: the home cage of the individual mouse could be placed at one end of the glass plate in the hope that the mouse will be motivated to run on the plate in order to reach its home cage; however, it may not be sufficient to motivate the animal to run with the highest velocity. This approach is based on anxiety, thus, making mice escape to a safe place (home cage). The question we are concerned with is how to increase mice’s compliance while preserving maximum animal performance in the most welfare-friendly way. Clicker training showed promising approaches to improving animal performance, as is reported in rats [6,7,8,9,10]; however, researchers reporting successful protocols did not describe them in detail [3,4,5,11,12,13,14]. From our experience, animal-friendly handling improves mild and moderate procedures in mice. For instance, we implemented positive reinforcement training to decrease anxiety-like behaviors, suggesting a less stressful experience for our mice and the experimenter [15].

Positive reinforcement training to improve the performance of mice in the CatWalk test may involve the implementation of the 3Rs principle (replace, reduce, and refine) while promoting animal welfare and scientific reproducibility, as required by Directive 2010/63/EU [16]. Refining with positive reinforcement requires, among others, the adoption of strategies to habituate and train animals to perceive fewer threats by gaining partial control over a situation with a reduction in stress [17,18]. Clicker training is a form of positive reinforcement in which expected behavior is compensated with a reward [15]. The researchers reported its successful application in companion, zoo, and laboratory animals [15,19,20]. As mice are notoriously fearful, clicker training of mice appears to be a genuine and accurate alternative to our laboratory routine [15]. Furthermore, factors that affect normal animal behavior, such as the experimenter and the environment, can subtly confound the experimental results [21], so training animals to achieve greater interaction with their experimenter and environment can improve welfare (by reducing stress) and science (by improving reproducibility) [22].

Currently, we apply clicker training protocols in mice to the CatWalk test with promising results. In this study, we evaluated the implementation of our clicker training protocol to improve participation in the CatWalk test in male and female mice. Subsequently, we evaluated the individual performance of mice on the Open Field (OF) and the Elevated Plus Maze (EPM) tests as indicators of stress and anxiety behaviors. We hypothesized that our clicker training protocol improved CatWalk performance and could potentially decrease the distress or anxiety caused in mice by the experimental setup and the experimenter.

## 2. Materials and Methods

This study was carried out following the ARRIVE guidelines [23]. The experimental design and management procedures were approved by the Rhineland-Palatinate State Authority (permit numbers: G-18-1-065) following the European Directive 2010/63/EU for the protection of animals used for scientific purposes.

### 2.1. Mice

Forty-eight C57BL/6JRj mice (24 eight-week-old males and 24 eight-week-old females) were purchased from a verified international breeder (Janvier Labs, Le Genest-Saint-Isle, France). All mice were raised following the recommendations of the Federation of Laboratory Animal Science Associations (FELASA). The mice were randomly housed in groups of four in type II long filter-top cages (Tecniplast, Buguggiate, Italy; SealSafe Plus, polyphenylsulfone, 36.5 cm × 21 cm × 14 cm), equipped with red transparent shelters, cocoons, and opaque 10 cm PVC tubes (tunnels) to transport the mice in an animal-friendly way. Housing followed a 12/12 light–dark cycle (200 lux from 7:00 to 19:00) in a temperature and humidity-controlled animal room (22–24 °C and 50–55%, respectively). Water and food (ssniff M-Z Extrudat, ssniff, Soest, Germany) were provided ad libitum. The mice were kept in same-sex groups of four. All animals were allowed a habituation period of one week before the experimental phase began.

### 2.2. Clicker Training Protocol

The present protocol was carried out over 11 days; 12 male and 12 female mice were clicker trained; the other 24 mice only received control handling. The clicker training protocol was conducted as previously reported [15]. Before the training period started, the reward (white chocolate cream) was placed into the home cages for 2–3 days. The same person carried out this protocol in a quiet room to reduce the stress of a new environment. The training lasted 5 min per mouse, divided into a series of 45 sec of training followed by a 15-second break during which all training equipment was removed from the cage. Each mouse was trained individually in the home cage, while the remaining group animals were transferred to a separate cage with their familiar enrichment. One training cycle was conducted per day. The reward was presented only for as long as it took the mouse to take one bite, except for the first display of learning a new task, which was rewarded with a “jackpot” reward (reward for three seconds).

Clicker training was established in sequential steps. A step was considered successfully learned after the animal showed 10 repetitions of the trained behavior within two minutes. Only after successfully training each mouse we moved to the next step. The clicker training protocol for the current experiment is depicted in Figure 1. In brief: (1) we established a conditioned connection between the “click” sound and the food reward by continuously clicking at the exact moment of white chocolate intake (4–5 s). This was repeated 3–4 times on day one; (2) we placed the familiar tunnel in the empty home cage and took the natural thigmotaxis of mice into account by placing it along a wall. As soon as the animal entered the tunnel, we clicked and presented the reward; (3) we placed a target stick (clicker device with an extendable arm and a small plastic ball attached) on one opening of the tunnel; when the mouse entered the other side, crossed the tunnel and touched the plastic ball with its nose, we administered the reward; (4) we placed the target stick close to a wall in the cage, and as soon as the animal touched it, we clicked, removed the stick, and rewarded. We proceeded by varying the position of the target stick in the cage (no tunnel was needed for this task); (5) As soon as each mouse touched the target stick, we started slowly moving the stick away from the mouse within the cage, leading the mouse to follow it. The first reward was given when following 1 cm; later, we extended the distance; (6) we transported the animals individually to the CatWalk and left them roaming freely for 1 min; (7) we repeated the target stick without tunnel on the CatWalk using the full length of the pathway without stopping in-between. Rewards were given at the two outer ends of the walkway immediately after the mouse walked the corridor.

### 2.3. CatWalk Test

The CatWalk XT (produced by Noldus Information Technology BV, Wageningen, The Netherlands) is a computerized gait analysis system for assessing forelimb–hindlimb coordination. The CatWalk structure is comprised of a 130 × 20 × 0.5 cm glass plate, a 120 × 5 cm plastic corridor (with no floor and ceiling) to narrow the running area on the glass plate, the moveable cover with inbuilt red light providing a background illumination for video acquisition, and a high-speed video camera mounted below the glass plate (Figure 2 and Figure 3a). A source of green LED light is mounted alongside the long edge of the glass plate in such a way that a green light enters the glass from the edge [24]. Thus, a green light is internally reflected inside the glass plate (the same principle is used in fiber optics technology). However, in those areas where the animal paws make contact with the glass plate, the green light is reflected at about 90° down and thus detected by the camera. The high-speed (100 frames per second) color camera captures these areas and sends the data to the CatWalk XT software. The red lamp, mounted on top of the mouse, provides good contrast between the paw prints and the rest of the body.

All 48 mice were moved to the neighboring test room in their home cages and habituated there for 1 h. In our experiment, each mouse underwent only one testing trial in the Catwalk. The light in the room was turned off, and after recording the background, one mouse was randomly selected among not yet tested individuals in its home cage, placed on one side of the corridor, and left undisturbed for the whole duration of the trial. The cover was closed, and the software recording started. During the recording period, the mouse was voluntarily moving back and forth in the corridor. Each time mouse moved through the recording area, a “run” was registered. The recording was stopped automatically after three compliant runs were reached. The trial was finished, and the mouse was placed back in its home cage. The software considers the runs to be compliant when the mouse moves from one side of the corridor to the other without hesitation, showing no rearing against the bounding walls or the side walls of the corridor, change in direction, straightening up on the bounding walls, or other substitute behaviors. A maximum speed variation of 60% was allowed. The number of runs to reach three compliant runs was recorded. Once each mouse accumulated three runs, data acquisition stopped automatically. Further, the running speed data was calculated. CatWalk XT software (Noldus Inc., Wageningen, The Netherlands) was used to analyze gait patterns. Before starting the next animal, the running path was cleaned with water.

### 2.4. Open Field (OF) Test

Running speed, wall latency, and time spent in the center and periphery of the OF test were evaluated in a white square plexiglass arena (dimensions: 40 × 40 × 40 cm) (Figure 4a). The mice were placed in the OF center (20 × 20 cm) and allowed to explore it for five minutes. Target behaviors were analyzed using the Ethovision XT software version 8.5.614 (Noldus Inc., Wageningen, The Netherlands). After each animal, the boxes were cleaned with water. No experimentalists were present in the room during the test.

### 2.5. Elevated Plus Maze Test (EPM)

The day after the OF test, the EPM was performed. The mice were placed in the central intersection (5 × 5 cm) from which the animal had free access to four arms (30 × 5 cm each) (Figure 5a). Two opposing arms were surrounded by opaque walls (15 cm), while the other two had no walls. The targeted behaviors were recorded for five minutes using an overhead video camera (ICD-49, Ikegami Electronics (Europe), Neuss, Germany) and analyzed by the Ethovision XT software version 8.5.614 (Noldus Inc., Wageningen, The Netherlands). The EPM was cleaned with water between animal tests. No mice were excluded from the test.

### 2.6. Statistical Analysis

Prior to statistical analysis, data were tested for normal distribution by D’Agostino and Pearson test. All the data shown in this manuscript were normally distributed. Clicker training data between males and females (Section 3.1) was evaluated by the student’s *t*-test. For all the other data (Section 3.2, Section 3.3 and Section 3.4), we used two-way ANOVA (main factors: training and sex) followed by Tukey’s Honest Significant Difference (Tukey HSD) test. Pearson’s correlation analyses were performed to establish relationships between clicker training and catwalk running speed. Statistical analyses were performed with GraphPad Prism, version 9.0, for Windows (GraphPad Software, San Diego, CA, USA). The F values indicated the variance ratio between and within groups. Degrees of freedom are shown as subscripted F values. Exact *p*-values were reported in the results and the figures. In all the figures, the values were expressed as means ± SD. We excluded one untrained female mouse from the OFT and one trained male mouse from the EPM test (open arms duration) based on the outlier Grubb’s test (using the log-normal correction). All staff involved in collecting data in the main study protocol were blinded whenever possible (e.g., video files and behavioral tests were analyzed by people not directly involved in the housing and training of mice).

## 3. Results

### 3.1. Clicker Training

Male and female mice participated well in training. Student *t*-test indicated that male mice showed slightly higher average repetitions per day of the desired behavior compared to female mice (Males: 5.7 ± 3.08; females: 7.8 ± 1.48 (mean ± SD); *n* = 12; t_22_ = 2.1; *p* = 0.02).

### 3.2. Performance on the CatWalk

We represent the scheme of the CatWalk test in Figure 3a. Both sexes showed higher values for running speed when trained compared to untrained (Figure 3b). The two-way ANOVA results indicated a significant effect of training and sex factors (Training F_1,44_ = 20, *p* ≤ 0.001; Sex F_1,44_ = 8.7, *p* = 0.005; Interaction F_1,42_ = 0.4, *p* = 0.52). Tukey HSD test revealed significant differences between trained and untrained animals regardless of sex (*p* = 0.04 and *p* = 0.004 for females and males, respectively). Furthermore, there was a significant relationship between the number of training repetitions per day (from Section 3.1.) and the running speed on the CatWalk (Pearson r = 0.48; *p* = 0.01; Appendix A).

Clicker training decreased the number of runs necessary to complete a CatWalk trial in both sexes (Training F_1,44_ = 7.3, *p* = 0.009; Sex F_1,44_ = 0.50, *p* = 0.48; Interaction F_1,44_ = 3.9, *p* = 0.05). Tukey HSD analyses showed a significant decrease in males but not in females compared to the untrained group (*p* < 0.01; Figure 3c).

**Figure 3 animals-12-03545-f003:**
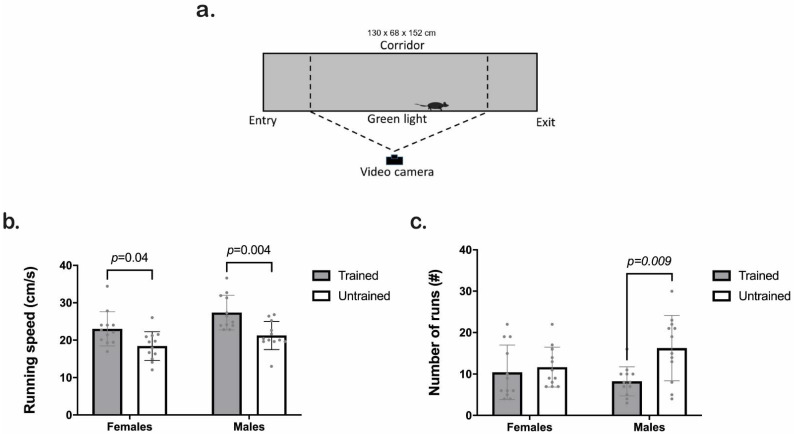
CatWalk XT performance of trained (gray) and untrained (white) C57BL/6J male and female mice. (**a**) Scheme of the CatWalk XT platform. (**b**) Running speed (cm/s). (**c**) Number of runs per group (#). We used two-way ANOVA followed by the Tukey HSD test for statistical analysis and multiple comparisons, *n* = 12. Each point represents an individual. Bars indicate the means ± SD. The exact *p*-value is provided when significant differences are given.

### 3.3. Open Field (OF) Test

We represent the scheme of the OF test in Figure 4a. Clicker training significantly increased the time spent in the center (Training F_1,43_ = 5.70, *p* = 0.02; Sex F_1,43_ = 0.71, *p* = 0.40; Interaction F_1,43_ = 1.6, *p* = 0.21). However, when comparing the groups by Tukey HSD test, we only observed a trend between trained and untrained females (*p* = 0.06; Figure 4b).

Clicker training significantly increased the distance traveled in the center (Training F_1,43_ = 15, *p* = 0.0004; Sex F_1,43_ = 0.08, *p* = 0.77; Interaction F_1,43_ = 1.1, *p* = 0.29). Tukey HSD test revealed specific differences between trained and untrained females (*p* = 0.01; Figure 4c), although such difference was not observed in males. Moreover, a subsequent evaluation of the ratio between distance traveled in the center and total distance traveled indicated a training factor effect (Training F_1,43_ = 8.6, *p* = 0.005; Sex F_1,43_ = 0.11, *p* = 0.74; Interaction F_1,43_ = 2.2, *p* = 0.14), but after a multiple comparison test, we found only a statistical trend between trained and untrained females (*p* = 0.06; Figure 4d).

**Figure 4 animals-12-03545-f004:**
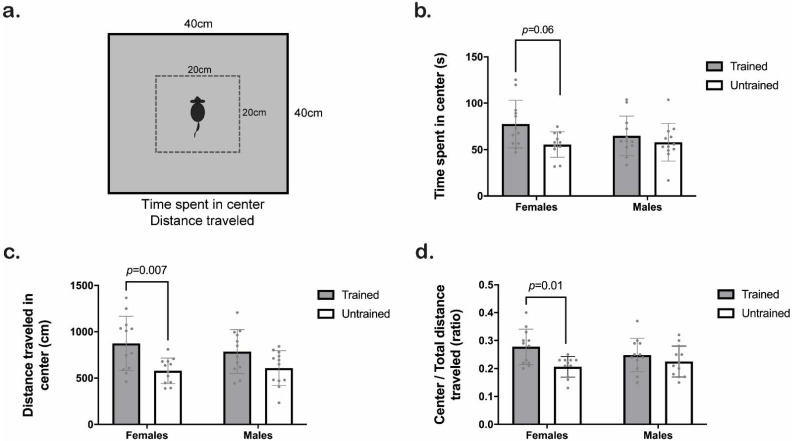
Behavioral activity of trained (gray) and untrained (white) C57BL/6J male and female mice in the open field (OF) test. (**a**) Schematic representation of the OF experiment. (**b**) Time the mice spent in the center (s). (**c**) Distance traveled in the center (cm). (**d**) Distance traveled in the center/distance traveled in total. Two-way ANOVA followed by the Tukey-HSD test was used for statistical analysis, *n* = 12 (untrained females *n* = 11). Each point represents an individual. Bars indicate the means ± SD. The exact *p*-value is provided when significant differences or trends are given.

### 3.4. Elevated Plus Maze (EPM)

We assessed specific parameters in the EPM test (running speed, distance traveled, and duration in open arms. Figure 5a). Clicker training increased the time mice spent in the open arms (Training F_1,43_ = 4.8, *p* = 0.03; Sex F_1,43_ = 0.74, *p* = 0.39; Interaction F_1,43_ = 0.23; *p* = 0.63), although Tukey HSD analyses showed no statistical differences (Figure 5b). The two-way ANOVA test from a distance traveled in the EPM revealed training and interaction effects (Training F_1,44_ = 17, *p* = 0.0002; Sex F_1,44_ = 0.41, *p* = 0.52; Interaction F_1,44_ = 7.4; *p* = 0.009). Tukey HSD analyses revealed a significant increase in the traveled distance of females (not males) when they were trained (*p* < 0.001; Figure 5c). Finally, trained mice had higher running speeds than untrained ones (Training F_1,44_ = 18; *p* = 0.0001; Sex F_1,44_ = 0.35; *p* = 0.55; Interaction F_1,44_ = 7.8; *p* = 0.007. Figure 5d). When assessing the number of entries in open arms (Appendix A), we found no statistical differences.

**Figure 5 animals-12-03545-f005:**
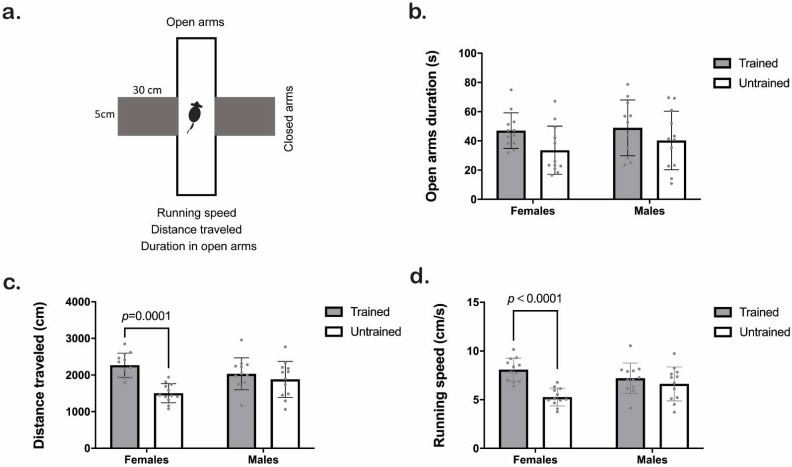
Behavioral activity of trained (gray) and untrained (white) C57BL/6J male and female mice in the Elevated Plus Maze (EPM) test. (**a**) Schematic representation of the EPM experiment. (**b**) Time spent in open arms (s). (**c**) Distance traveled in the EPM (cm). (**d**) Running speed (cm/s). We used Two-way ANOVA followed by the Tukey HSD test for statistical analysis, *n* = 12 (trained males in 5b *n* = 11). Each point represents an individual. Bars indicate the means ± SD. The exact *p*-value is provided when significant differences are given. Two-way ANOVA analysis for open arm duration (**b**) indicated an effect of training, although the Tukey HSD test did not show differences.

## 4. Discussion

We demonstrated for the first time that clicker training improved the performance of mice in the CatWalk XT test.

Clicker training started in the home cage before transferring to the CatWalk, simplifying the learning process while maintaining a familiar environment and a parallel reduction in stress factors. In other words, we simultaneously trained the mice in the CatWalk as habituation. We found that clicker-trained mice increased their running speed and decreased the number of runs. The average speed of mice crossing the corridor increased, suggesting that they increased their motivation to obtain the reward and thus ran faster to the exit at the end of the corridor. The improvement in motivation is a powerful indicator that our clicker-trained mice crossed the corridor without hesitation.

Our findings indicated that clicker training lowered the number of runs to complete the CatWalk. It means that clicker training may offer promising alternatives for both mice and experimenters. On one side, animals are not forced to cross the corridor (e.g., by puffing them), improving well-being along the test. On the other side, even if training requires a time investment, it reduces the number of runs per mouse, optimizing the time required by the experimenter to assess each mouse in the CatWalk test. We can further hypothesize that our protocol has the potential to decrease animal numbers by reducing non-compliant individuals or animals failing to meet run criteria. The fact that we identified clicker training was adequate for training mice for CatWalk challenges led to the assumption that, unlike rats, mice cannot be trained to make uninterrupted runs [25]. In this study, we did not evaluate specific gait parameters, but we hypothesize that clicker training would improve mice performance in the CatWalk test. We are addressing such a topic in our current research.

As mentioned, clicker training increased running speed in the CatWalk. Previous experiments reported changes in the running speed of rats when the reward changed either in quality or quantity, indicating a negative contrast effect [26,27]. Furthermore, studies that incorporated self-selection of music (a potential reward) during exercise increased running speed and general performance in humans [28,29]. In this regard, we must be careful not to humanize mouse data, but our findings may suggest that positive reinforcement promoted increases in running speed, suggesting positive welfare statuses.

We observed a significant effect of training in running speed and distance traveled on the CatWalk, OF, and EMP. Differences were significant in female (trained vs. untrained) but not in male mice. A recent study demonstrated that the gait performance of young mice assessed by CatWalk depended on age and sex, suggesting that sex hormones and genes on the X and Y chromosomes may impact behavioral outcomes [30]. Konhilas et al. (2004) [31] argued that females had higher aerobic capacity than males, most likely due to intrinsic differences in heart and skeletal muscle. Furthermore, differences in behavior depend on the hormonal status of young adult mice [32]. In addition, ovariectomized mice and rats significantly reduced wheel running compared to their non-ovariectomized and ovariectomized estrogen-receiving counterparts [31,32]. Although we found no substantial differences between females and males, we may speculate that positive reinforcement may modulate sex-intrinsic traits.

Our assumption that training may be beneficial for the well-being relies on the fact that when trained (compared to those not trained), mice increased the distance traveled on the EPM and the OF tests while showing a strong tendency to stay in the center of the OF. When comparing within sex, we found that female mice were significantly more influenced by training than male mice.

We can infer that clicker training exerted potential reductions in female mice’s stress and anxiety-like behaviors due to their tendency to explore and interact with the stimuli [33]. Evidently, the physiological and hormonal status may play a sex-specific role in these behaviors, making female mice more susceptible to clicker training. However, it is a matter of further research.

According to our observations, an unexpected finding was the lower behavioral response between trained and untrained males (compared with trained and untrained females). We were surprised because the males performed better in home cage training and required fewer runs to comply with the CatWalk test. Based on our qualitative observations, an explanation of these findings was that separating the males from the group and putting the animals back together (after training or conducting a test) resulted in fights, which could have caused stress to the male animals.

Our present results indicate that clicker training can improve performance in the CatWalk test and may positively influence mouse welfare. However, we acknowledge the limitation of the absence of stimuli on untrained animals so that we can directly compare puffing vs. clicker training, which is a matter of our current experiments.

## 5. Conclusions

Male and female mice benefited from clicker training. From the point of view of welfare, the cooperation of animals with the clicker training protocol emerges as a promising way to reduce stress and anxiety-like behaviors. However, we recommend more research on the potential influence of clicker training (such as environmental enrichment) on specific experimental arrangements. We also recommend additional studies modulating the frequency, quality, and quantity of rewards in male and female mice to establish the benchmark by which we improve welfare without threatening the reproducibility of the results.

## Figures and Tables

**Figure 1 animals-12-03545-f001:**
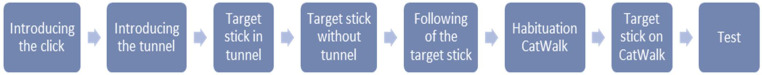
Graphical illustration of the clicker training protocol.

**Figure 2 animals-12-03545-f002:**
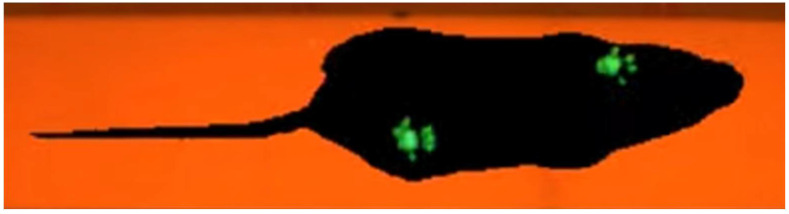
Mouse performing the Catwalk test. The footprint of each paw is visualized in green color.

## Data Availability

The data presented in this study are openly available in FigShare at doi 10.6084/m9.figshare.21719957.

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
