# Peer review of "Clicker Training Mice for Improved Compliance in the Catwalk Test"

_animals, 2022, doi:10.3390/ani12243545_

Round 1

Reviewer 1 Report

The study evaluates the use of clicker training to replace an aversive practice in mice in a locomotion test called catwalk. Additionally, they tested the trained mice versus the untrained in an open field chamber and a elevated plus maze test. Results indicated that clicker training improved running speed and lowered anxiety-like behaviors.

Introduction

Reference 15 and 16 are exchanged, please verify each one in the text. 

Materials and methods

They mention offering chocolate as a reinforcer, 3-4 times a day, this suggests a significant amount that could be harmful to the health of the mice. Please indicate the amount in grams if you have the data, and comment on it in the discussion if it represents a risk or if the amount of chocolate is safe for the health and welfare of the animals.

L138 repeats information mentioned in L107, please improve it.

Please clarify the chronological order of the tests, how many days passed between the catwalk, the OF and the plus maze. It is mentioned that EPM occurred the next day of the OF, but it is not clear when mice performed OF in relation to the catwalk test.

L179 Authors consider values between P>0.05 and ≤0.1 as favorable statistical trends, please justify. Traditionally only significant differences are validated and discussed.

L139 Please clarify the meaning of habituated in this idea. How long they remained in test, were they exposed to the test previously?

Mention if food was used during test, if no trained animals were exposed or habituated to the catwalk the same time as the trained animals before test, or if test was the first time untrained mice were placed there?

Discusion 

It seems as clicker device was unecesary, please discuss why not simply use positive reinforcement, for instance simply food instead of create a bridge (clicker device) and spent time creating an association between clicker sound (neutral stimulus) and food (unconditioned reinforcement) 

Author Response

Dear reviewer, thank you very much for taking the time to assess and improve our manuscript. We also thank you for your very constructive comments.

We marked all changes in the manuscript (track changes) and answered comments in the below point-to-point reply.

  1. Reference 15 and 16 are exchanged, please verify each one in the text. 

Thank you very much for your comment. We apologize for the mistake. At the moment of uploading the manuscript, reference “1” was duplicated, distorting all other references. We fixed this issue.

  1. They mention offering chocolate as a reinforcer, 3-4 times a day, this suggests a significant amount that could be harmful to the health of the mice. Please indicate the amount in grams if you have the data, and comment on it in the discussion if it represents a risk or if the amount of chocolate is safe for the health and welfare of the animals.

Thank you very much for this comment. We provided to each animal with just a short lick of the white chocolate cream. White chocolate contains negligible amounts of theobromine; therefore, the health risk is quite low. We believe that a low amount of eaten cream does not compromise glucose levels. However, it will be matter of our present research work.

  1. L138 repeats information mentioned in L107, please improve it.

We acknowledge you pointed to this redundancy. We deleted the text from line 138.

  1. Please clarify the chronological order of the tests, how many days passed between the catwalk, the OF and the plus maze. It is mentioned that EPM occurred the next day of the OF, but it is not clear when mice performed OF in relation to the catwalk test.

We thank your comment. What is meant is, that we implemented each behavioral test per day, always the following day.

  1. L179 Authors consider values between P>0.05 and ≤0.1 as favorable statistical trends, please justify. Traditionally only significant differences are validated and discussed.

We thank you for your useful comment. We deleted this statement as it may give a misleading impression.

  1. L139 Please clarify the meaning of habituated in this idea. How long they remained in test, were they exposed to the test previously?

Habituation here means the acclimatization of mice to the training room after the transfer from the housing room. The CatWalk test took approximately 20-40 minutes. Before the real test, animals were already exposed to the CatWalk device two times, as shown in the clicker training protocol section 2.2.  

  1. Mention if food was used during test, if no trained animals were exposed or habituated to the catwalk the same time as the trained animals before test, or if test was the first time untrained mice were placed there?

Thank you very much, dear reviewer. During the CatWalk test, food was not presented. Untrained animals were habituated to the CatWalk for the same amount of time as the trained ones.

  1. It seems as clicker device was unecesary, please discuss why not simply use positive reinforcement, for instance simply food instead of create a bridge (clicker device) and spent time creating an association between clicker sound (neutral stimulus) and food (unconditioned reinforcement) 

Thank you for your meaningful comment. We agree that following the target stick could be replaced with a spatula with chocolate with positive results, too. However, we have built up the behavior in very small steps and we have made good experiences with the clicker training with the small often very scurrying mice. The risk to scare the animals with fast movements with the only food reward, or that you simply cannot reward fast and precise enough is quite high, which could cause additional noise to the experiment. Nevertheless, we are constantly working on improving our techniques and this might be an option to consider.

Reviewer 2 Report

Interesting study describing a nice refinement technique for mice assessed on the CatWalk device.

In section 2.1:  How did you determine that 12 mice per group and sex were required to achieve statistical significance?  Especially given the "trending" data that you report later, did you perform a power analysis prior to initiating the study?

Also in section 2.1:  How did you randomize them?  How did you determine order to perform on the CatWalk and to do the training?  Could the training/testing order affect the results?

Section 2.4:  You noted that outliers were excluded.  How many animals did this represent?  Were they replaced?

Section 2.6:  You are reporting error bars in your graphs.  Are these standard deviation or standard error?

Section 3.2:  The first sentence says this is the OF, but this figure appears to be the CatWalk?

Discussion, lines 264-266:  Please be sure to address that, although the training results in less time on the CatWalk device, there is still time that the investigator must invest in training the mice.  In no way am I indicating that it is time wasted, but as presented now, it implies labor savings, which is not quite correct.

Author Response

Dear reviewer, thank you very much for taking the time to assess and improve our manuscript. We also thank you for your very constructive comments.

We marked all changes in the manuscript (track changes) and answered comments in the below point-to-point reply.

  1. In section 2.1:  How did you determine that 12 mice per group and sex were required to achieve statistical significance?  Especially given the "trending" data that you report later, did you perform a power analysis prior to initiating the study?

Thank you very much for such an important comment. We planned the study as a “proof of principle”, this is why the classical power analysis was not performed. Our estimation of group size was based on the available literature in CatWalk and consultations with experts in the field.

  1. Also in section 2.1:  How did you randomize them?  How did you determine the order to perform on the CatWalk and to do the training?  Could the training/testing order affect the results?

Thank you for your comment. Indeed, testing order can affect behavioral results. As our interest was to evaluate the effect of clicker training in the CatWalk test, the sequence of tests was Clicker Training-CatWalk-OFT-EPM.

Animals for our experiment were commercially supplied (Janvier Labs, Le Genest-Saint-Isle, France). After delivery, mice were randomly picked from the transportation cage and placed into home cages.

  1. Section 2.4:  You noted that outliers were excluded.  How many animals did this represent?  Were they replaced?

Thank you very much for pointing this out. We excluded one untrained female mouse from the OFT and one trained male animal from the EPM based on the outlier Grubb´s test (using the log-normal correction). Please find this information in section 2.6. (Statistical Analysis).

  1. Section 2.6:  You are reporting error bars in your graphs.  Are these standard deviation or standard error?

Thank you very much for your comment. We are reporting mean ± SD (standard deviation). This information is provided in section 2.6. (Statistical Analysis) and in each figure legend.

  1. Section 3.2:  The first sentence says this is the OF, but this figure appears to be the CatWalk?

Thank you for pointing out this mistake. It was corrected.

  1. Discussion, lines 264-266:  Please be sure to address that, although the training results in less time on the CatWalk device, there is still time that the investigator must invest in training the mice.  In no way am I indicating that it is time wasted, but as presented now, it implies labor savings, which is not quite correct.

We thank you for this insightful comment. We agree on training cost time and we modified the text in the manuscript.  Moreover, it is worth mentioning that clicker training reduced the number of runs per mouse, optimizing the time required by the experimenter during the potentially stressful assessment during the CatWalk test. 

Reviewer 3 Report

The present manuscript describes a clicker training procedure that improves the quality of measurements derived from the CatWalk test, using mice as subject. The experimental design consisted of two factors: Sex (males versus females), Training (untrained versus clicker-trained mice) and their interactions were analyzed. The results were in the hypothesized direction, namely that clicker training improved data quality in the CatWalk task. The effects of the training procedure on stress and anxiety was in addition tested in the Open field and the Elevated plus-maze. Sometimes, the authors suggest that these  two additional tests yield an indication of the animal’s well-being and that the reduction in the number of runs and the higher running speed might indicate a better well-being. I would be more cautious to draw this conclusion about well-being. Some results appeared to be sex dependent. The authors performed post hoc comparisons between the four testing by sex groups, even when the two factorial ANOVA didn’t yield an interaction effect.

The study makes sense and appears to be well performed. However, considering that the manuscript’s topic is a refinement of a testing procedure, I miss a large number of details concerning the procedures followed for testing, and about the statistics applied. Also, the authors should address the possibility that clicker training can act as environmental enrichment and should discuss its potential effects on subsequent testing.

The numbering of the references in the manuscript needs to be corrected.

Specific comments

1) Lines 11-12: Use runway or corridor instead of platform

2) Line 44: “Individual animals must run on a glass plate …” Please check in the entire manuscript: mice are required to run, not walk (as stated in the abstract)

3) Line 66: “(…) test must involve the implementation of the 3R principle …” Positive reinforcement does not implement the 3Rs, but only refines the CatWalk procedure.

4) Lines 76-77: (…) so training animals to achieve greater interaction with their surroundings can improve welfare and science…” This statement needs more explanation. Now it is too general.

5) Line 80: “(…) to improve voluntary participation in the CatWalk test…” I doubt that the mice participate voluntarily. They are put into the apparatus by the experimenter and are required to cross the runway.

6) Lines 96-97 and 102-103: “The mice were kept in same-sex groups of four animals.” In lines 102-103, it is stated that the mice were kept in groups of 4, whereas in lines 96-97, it is state that the mice were housed in groups of up to four animals, suggesting that some groups must have been smaller than four animals.

7) Lines 105-120: This is a very poor description of the procedure to assign mice to the two different conditions, of the "training cage", of the handling involved in training, the contingencies that were in effect, the mode and exact time point of presentation of the chocolate reward, etc.... The present description fails to provide the details necessary for replicating this study.

8) Line 106: “(…) as previously reported [15]… I didn't check, but I doubt that Directive 2010/63/EU contains a description of the clicker training procedure.

9) Line 112: “(…) 10 repetitions of the trained behavior within 2 minutes.” What exactly did the mice learn? What contingency was in effect? The click was given whenever the mouse took a piece of chocolate. How was the chocolate presented? How large (weight) was the chocolate reward?

10) Lines 113-114: “The initial phase of training took place in the training cage where each entry into the already familiar tunnel was rewarded.” What is the training cage (describe in detail)? Were mice individually transferred to the training cage?

11) Line 115: “(…) to cross through the tunnel and touch the target stick in…” Which tunnel? What is a "target stick"? How and when exactly was the reward (chocolate?) presented? (Note: for these important details, it isn’t sufficient to refer to a JOVE publication.)

12) Lines 117-118: “Once the animals followed the target stick through the home cage, …” The training cage thus is the home cage? What happened with the other three inhabitants of the cage during training of a mouse?

12) Lines 118-119: “(…) they were habituated to the catwalk environment on the following training day and subsequently clicker trained there.” Again, provide details about the procedure applied.

13) Line 124: “(…) a 130 x 20 cm glass plate,…” I would call this a runway or a corridor with a floor of glass. Also report how narrow the runway was made for mouse testing with the inserted plastic side walls.

14) Line 126: (…) (Figure 1)”. Also refer to Fig. 2a here.

15) Line 139: “The mice were moved and habituated to the test room.” I assume that the mice were transported in their home cages to the testing room. How and how long were they habituated? How long were they housed in the testing room before testing started?

16) Lines 140-141:  “(…) after recording the background, a mouse was placed on one side of the corridor.” The home cage contained 4 animals. How were the animals in a cage selected for testing? Were tested animals put back in the home cage directly after testing? Please provide sufficient detail so that other researchers are able to replicate your study.

17) Line 141: “The cover was closed (…)” Which cover?

18) Line 144: “(…) showing no change in direction, straightening up on the bounding walls,…” Do you mean:  rearing against the bounding walls or the side walls of the corridor?

19) Lines 145-146: “The number of runs to reach three compliant runs was recorded.“ Please describe in detail how the mice were handled between trials, and at the end of a testing session.

20) Lines 149-150: “After each mouse was tested, the running path was cleaned with water.” Do you mean: after a mouse had completed its runs, the runway was cleaned with water. Now it reads as if the runway was only cleaned once, after the last tested mouse.

21) Line 153: “(…) white square plexiglass arena …” Please refer to Fig. 3a.

22) Lines 158-159: “Outliers (Iterative Grubbs; Alpha = 0.2) were excluded if confirmed to be due to errors.” It is a dangerous practice to routinely perform outlier tests. What errors exactly were made that warranted exclusion of obtained data? How many valid data were obtained (or: how many runs/mice were excluded from analysis)?

23) Lines 162-163: “(…) (30 x 5 cm each).” Please refer to Fig. 4a here.

24) (…) by D'Agostino & 169 Pearson, Shapiro-Wilk, Kolmogorov-Smirnov, and Anderson-Darling tests.” Lines 169-170: These test have a different sensitivity to detect deviations from the normal distribution. Which criterion exactly was used? Or did you require that the data were normally distributed if all test performed confirmed normality?

25) Lines 170-171:  “Normal data were compared using the student's t-test or two-way ANOVA (main factors: training and sex)…” how were variables analyzed that were not normally distributed?

26) Line 172: “(…) followed by Tukey's Honest Significant Difference (Tukey-HSD) test.” I'm confused: a t-test compares two groups. There is no need for post hoc comparisons. Did you compare the four training/sex groups if the ANOVA revealed a training by sex interaction routinely by post-hoc comparisons?

27) Lines 172-174: “Pearson's and Spearman's rank correlation analyzes were performed to establish relationships between clicker training and catwalk running speed.” Which samples were used for the correlations? Why were both rank correlation coefficients calculated?

28) Lines 177-178: “The level of significance for all tests was established as significant P≤0.05 (*) and highly significant P≤0.01 (**) or P≤0.001 (***).” This practice doesn't make much sense. See e.g. DOI 10.1007/s10654-016-0149-3 and DOI: 10.1098/rsbl.2019.0174

29) Lines 179-180: “Values are expressed as means.” Add that the SDs were reported together with the means.

30) Line 180: “Individuals considered outliers (Iterative Grubbs; Alpha = 0.2)…” What is meant by "individuals"? The mouse, results of a particular test or individual test values? All relevant details of the statistics performed should have been given in detail in section 2.6. Statistical Analysis.

31) Lines 187-188: “(…) (Males: 5.7, females: 7.8;…” Add an index for variation (SEM, SD, confidence interval).

31) Lines 212-214: “The ratio between time spent in the center/total time spent in the OF test did not differ statistically (Supplementary figure 2).” Why did you analyze the ratio (the time in center and at border are mutually exclusive, and the open field test duration was 5 minutes. In that case either the time in center or the time at border suffices for analysis.

For distance traveled, this is different, as the activity in the center and at the borders might be very different.

32) Line 225: “(a) Schematic representation of the OF experiment.” Please check the drawing of the open field. The position of the outer walls appear to have shifted upwards.

33) Lines 232-233: “(…) the time mice lasted in the open arms…” I suggest to replace “lasted” by “spent”.

34) Line 234: “(…) Tukey HSD analyses showed no statistical differences (Figure 4b).” I'm not surprised: there is no differential effect of training by sex (no interaction) and no sex effect. The factor sex is a comparison of males vs. females across the two test conditions, and the comparison has more DFs than the four individual post hoc comparisons where the p-value is also corrected for multiple comparisons.

34) Lines 246-247: “Each point represents an individual” Add a full stop at the end of this sentence: “Each point represents an individual.”

35: Line 264: (…) improving relative well-being along the test.” What is "relative well-being"?

36: Line 270: “(…) [25].” I was unable to find this assumption in the indicated reference. I'm afraid that the numbering of references in the manuscript is incorrect (check whether you deleted or added a reference and forgot to update the reference list accordingly before submitting the paper). See also my comment to the reference in line 106.

37): Lines 278-279: “(…) our findings may suggest that running is speed incentivized by positive reinforcement indicates positive welfare statuses.” Please rephrase this sentence and improve its readability.

38) Lines 305-308: “Based on our qualitative observations, an explanation of these findings was that separating the males from the group and putting the animals back (after training or conducting a test) resulted in fights, which could have caused stress to the male animals.” The order of testing and putting back tested mice might affect the stress response of the other inhabitants of a cage: https://doi.org/10.1016/j.physbeh.2009.03.008 This may affect the subsequent testing of these animals.

39) Line 313: “(…) floor targets,…” "floor targets" being what?

40) Line 317: “(…) the voluntary cooperation of animals…”  I don't understand how this study shows the the animals cooperated voluntarily.

41) Lines 318-319: “However, researchers must be careful of potential clicker training interference,…” Note that training may act as environmental enrichment which may affect behavior in the CatWalk test, but also in other tests that are eventually performed after the CatWalk test.

42) Line 339 ff. (References): Please add the DOI numbers to the references. Update the reference list so that it matches the reference numbering in the manuscript.

Round 2

Reviewer 3 Report

The authors have appropriately and carefully revised their manuscript.